# Current Therapeutic Strategies and Prospects for EGFR Mutation-Positive Lung Cancer Based on the Mechanisms Underlying Drug Resistance

**DOI:** 10.3390/cells10113192

**Published:** 2021-11-16

**Authors:** Yukari Tsubata, Ryosuke Tanino, Takeshi Isobe

**Affiliations:** Division of Medical Oncology & Respiratory Medicine, Department of Internal Medicine, Shimane University Faculty of Medicine, 89-1 Enya-Cho, Izumo 693-8501, Japan; rtanino@med.shimane-u.ac.jp (R.T.); isobeti@med.shimane-u.ac.jp (T.I.)

**Keywords:** epidermal growth factor receptor gene mutation, epidermal growth factor receptor-tyrosine kinase inhibitor, acquired resistance, pemetrexed, targeted therapy

## Abstract

The discovery of activating mutations in the epidermal growth factor receptor (*EGFR*) gene and the development of EGFR tyrosine kinase inhibitors (TKIs) have led to a paradigm shift in the treatment of non-small cell lung cancer (NSCLC). *EGFR* mutation-positive NSCLC is common in East Asia, and approximately 50% of adenocarcinomas harbor *EGFR* mutations. Undoubtedly, EGFR-TKIs, with their promising efficacy, are the mainstay of primary therapy. However, even if tumor shrinkage is achieved, most patients become resistant to EGFR-TKIs and relapse; hence, EGFR-TKIs do not achieve a radical cure. The problem of the development of resistance to targeted drugs has been a persistent challenge. After the role of *EGFR* T790M mutation in acquired drug resistance was reported, osimertinib, a third-generation irreversible EGFR-TKI, was designed to overcome the resistance conferred by T790M mutation. In addition, some studies have reported the mechanism of drug resistance caused by mutations other than the T790M mutation and strategies to overcome them. Elucidating the mechanism underlying drug resistance development and combining therapeutic approaches are expected to further improve NSCLC prognosis.

## 1. Introduction

The role of activating mutations in the epidermal growth factor receptor (*EGFR*) gene in non-small cell lung cancer (NSCLC) was reported in 2004 [1]. The development of EGFR tyrosine kinase inhibitor (TKI), as the therapeutic agent, radically overturned the conventional chemotherapy regimen for lung cancer, which was treated with cytotoxic anticancer agents until then [2,3]. Thus, the conventional method of synthesizing numerous candidate compounds and selecting an effective anticancer drug from among them has been replaced by a method of finding a target molecule first and then designing and developing anticancer drugs that specifically inhibit the target molecule. Since then, several molecular targeted drugs have been developed and clinically applied in patients harboring *ALK* fusion genes, *ROS1* fusion genes, *BRAF* mutations, *NTRK* fusion genes, and *MET* mutations [4,5,6,7,8]. Thus, personalized treatment based on the type of genetic mutations harbored by the patients is the new mainstay of chemotherapy for NSCLC.

Genes whose mutations and alterations are directly responsible for the increased growth and progression of cancer are collectively referred to as driver genes. Kinase inhibitors targeting these mutations/alternations are the primary therapeutic choices for patients harboring driver gene mutations. However, the development of drug resistance is one of the most common challenges associated with the use of kinase inhibitors. Patients treated with kinase inhibitors following fulfilling eligibility criteria show remarkable tumor shrinkage; however, despite the high efficacy of these kinase inhibitors, radical cure of stage IV NSCLC remains an unachieved goal because invariably, almost all patients develop acquired resistance within one to several years. Elucidating drug resistance mechanisms and developing new drugs to overcome resistance is indispensable for achieving higher efficacy and radical cancer cure. Osimertinib was the first drug developed to overcome resistance to kinase inhibitors. A third-generation irreversible EGFR-TKI, osimertinib overcomes the drug resistance acquired due to *EGFR* T790M mutation [9]. Since then, constant efforts have been made to develop drugs for overcoming resistance to kinase inhibitors and cytotoxic anticancer agents. New molecular targeted drugs and their combination with cytotoxic anticancer agents are expected to improve the prognosis of NSCLC patients further. This review focuses specifically on the latest treatments for *EGFR* mutation-positive lung cancer and the strategies aimed at overcoming drug resistance; moreover, we discuss the prospects for their use in clinical settings.

## 2. Advances in the Treatment of *EGFR* Mutation-Positive Lung Cancer

### 2.1. First and Second-Generation EGFR-TKI Monotherapy

*EGFR* mutations are found in approximately 15% of lung adenocarcinomas cases in Europe and the United States, and 55% cases in East Asia [10]. The exon 19 deletion mutation and L858R mutation account for more than 90% of *EGFR* mutations, making tumors highly sensitive to EGFR-TKIs [11]. Other mutations are rather uncommon, and a few among them, such as G719X, L861Q, and S768I, turn tumors highly sensitive to EGFR-TKIs; in NSCLC patients harboring these mutations, EGFR-TKIs are considered as the main therapeutic agents. On the contrary, patients harboring exon 20 insertion mutations or T790M mutations as primary mutations are resistant to EGFR-TKI and are not indicated for treatment [12].

Phase III clinical trials comparing the use of EGFR-TKIs (gefitinib and erlotinib) with platinum in NSCLC harboring a deletion mutation in exon 19 and the L858R mutation in exon 21 [2,3,13,14] showed that EGFR-TKI monotherapy significantly improved the response rate and progression-free survival (PFS) compared to platinum combination therapy, establishing it as the standard of care (Table 1). Compared to EGFR-TKIs, phase-III clinical trials comparing afatinib (LUX-Lung7) and dacomitinib (ARCHER1050) (both second-generation EGFR-TKIs) with gefitinib (a first-generation EGFR-TKI) [15,16] reported that PFS (the primary endpoint) was 13.0 months vs. 10.4 months (HR 0.81, 95% C.I. 0.62–1.05) for LUX-Lung7 and 14.7 months vs. 9.2 months (HR 0.59, 95% C.I. 0.47–0.74) for ARCHER1050; in both the cases, the second-generation EGFR-TKI group showed superior outcomes (Table 1). Based on the results of these pivotal clinical trials, both first- and second-generation EGFR-TKIs are being used in clinical settings worldwide.

### 2.2. Treatment of EGFR T790M Acquired Resistance

Osimertinib is an irreversible EGFR inhibitor designed to inhibit both EGFR-sensitive and T790M-resistant mutations and is classified as a third-generation EGFR-TKI. In the AURA3 study, a phase III trial comparing osimertinib with platinum-based combination therapy in patients who tested positive for the T790M resistance mutation after treatment with first- and second-generation EGFR-TKIs [9], the median PFS (the primary endpoint) significantly prolonged to 10.1 months vs. 4.4 months (HR 0.30, 95% CI 0.23–0.41). The frequency of grade 3 or higher toxicity was lower with osimertinib. Therefore, if a therapeutic agent other than osimertinib is chosen for the first-line therapy, the *EGFR* mutation status should be rechecked after disease progression, and if the T790M mutation is detected, osimertinib use should be strongly considered.

### 2.3. Third-Generation EGFR-TKI Monotherapy

The FLAURA study, a phase-III trial comparing single-agent osimertinib with first-generation EGFR-TKIs (gefitinib or erlotinib) in patients with *EGFR* mutations (exon 19 deletion mutation and exon 21 L858R mutation) [17], reported that PFS (the primary endpoint) significantly prolonged with osimertinib treatment, 18.9 months vs. 10.2 months (HR 0.46, 95% C.I. 0.37–0.57). Overall survival (OS) also significantly improved with osimertinib, 38.6 months vs. 31.8 months (HR 0.799, 95% C.I. 0.641–0.997). In terms of toxicity such as skin rash and liver damage, osimertinib tends to be milder than first-generation EGFR-TKIs, and osimertinib monotherapy has been established as one of the standard therapies for the first-line treatment of *EGFR* mutation-positive NSCLC (Table 1).

Meanwhile, several clinical trials have begun exploring the usefulness of osimertinib combination therapy. Combination therapy of osimertinib and CBDCA/PEM in FLAURA2, an international safety run study, exhibited a manageable safety and tolerability profile in first-line settings [18]. In addition, in phase II clinical trial (WJOG9717L) examining the efficacy of bevacizumab (an angiogenesis inhibitor) and osimertinib, the primary endpoint, PFS, did not show a statistically significant prolongation of survival [19]. Further assessment is expected.

### 2.4. EGFR-TKI Combination Therapy

Combination therapies using EGFR-TKIs, and cytotoxic agents or angiogenesis inhibitors are also being explored (Table 2). Two phase-III trials compared the patient outcomes following treatment with gefitinib alone with that in response to gefitinib plus carboplatin plus pemetrexed (PEM) in combination with cytotoxic agents. The NEJ009 trial [20] showed improved PFS (the primary endpoint) in the combination group (20.9 months vs. 11.9 months, HR 0.490, 95% C.I. 0.39–0.62). The median OS in the combination group was an astounding 50.9 months, indicating that patients with *EGFR* mutation-positive lung cancer may now have 5-year survival even after being diagnosed with unresectable lung cancer. A study conducted by Noronha et al. also showed prolongation of the primary endpoints, PFS, and OS [21].

Outcomes of combination therapy with erlotinib and anti-angiogenesis agents have also been evaluated in several trials. The NEJ026 trial [22,23] was a phase III trial that compared erlotinib monotherapy with erlotinib plus bevacizumab; PFS, the primary endpoint, was prolonged in the combination therapy group (16.9 months vs. 13.3 months, HR 0.605, 95% C.I. 0.417–0.877), but OS did not improve. The RELAY trial [24] was a phase III trial comparing erlotinib alone with erlotinib plus ramucirumab and the PFS (primary endpoint) prolonged in the combination group (19.4 months vs. 12.4 months, HR 0.59, 95% C.I. 0.46–0.79).

It remains inconclusive whether starting the first-line treatment of *EGFR* mutation-positive lung cancer with osimertinib alone, with other EGFR-TKIs alone, or using combination treatment first and then using osimertinib after T790M is detected, improves the prognosis. A study to confirm the efficacy of osimertinib plus a cytotoxic anticancer agent (NCT04035486) and a clinical trial comparing osimertinib with erlotinib plus ramucirumab in L858R mutation-positive patients only (JapicCTI-184146) are ongoing, and their results are awaited.

### 2.5. Immunotherapy for EGFR-Mutated Lung Cancer

These days, immune checkpoint inhibitors (ICIs) (including anti-PD-1/L1 and anti-CTLA-4 antibodies) are an essential therapeutic option in the treatment of lung cancer. *EGFR* activation may increase PD-L1 expression in tumor cells by upregulating PI3K and activating the IL-6/JAK/STAT3 pathway and MEK/ERK pathway [25,26,27,28]. It has also been reported that the administration of EGFR-TKIs has a positive effect on the tumor microenvironment [29,30,31], but the specific molecules involved in PD-L1 activation have not yet been elucidated. Furthermore, the correlation between the presence of *EGFR* mutations and PD-L1 expression is still controversial. However, the intensity and treatment susceptibility of PD-L1 expression in T790M-negative patients are consistently correlated, regardless of prior treatment with EGFR-TKIs [32,33]. This indicates that elucidating the relationship between PD-L1 expression and EGFR-TKI susceptibility, as well as the resistance mechanisms, may provide new avenues for overcoming EGFR-TKI resistance.

On the other hand, the therapeutic effect of combining ICIs and chemotherapy for treatment after EGFR-TKI failure remains unclear because *EGFR*-positive lung cancer has been excluded from many clinical trials [34]. The IMpower150 study group examined the effect of adding atezolizumab to CBDCA/PTX/BEV in chemotherapy-naive non-squamous NSCLC and allowed registration of *EGFR*-positive lung cancer after EGFR-TKI failure. A subgroup analysis of this cohort revealed useful results for median OS in the atezolizumab-added group (not evaluated vs. 17.5 months, HR 0.31) [35]. However, the therapeutic effect of ICIs alone for *EGFR*-positive lung cancer is limited, and the PFS of nivolumab monotherapy after EGFR-TKI failure is only 1.7 months [36]. It is presumed that such a gap between the results of preclinical studies and actual clinical trials is due to various factors, such as tumor heterogeneity, PD-L1 IHC scoring criteria, and treatment history. Because some *EGFR* mutation-positive patients may benefit from immunotherapy, it is important to elucidate the molecules primarily involved in the mechanism of PD-L1 activation and EGFR-TKI resistance and to investigate their association. Identifying predictors of therapeutic efficacy is considered to be an important element of personalized treatment.

## 3. Mechanisms Underlying Acquired Resistance to EGFR-TKIs

Broadly three mechanisms of acquired resistance to EGFR-TKIs have been reported [37]: (1) alteration of the target via EGFR-TKI susceptibility mutations (acquiring T790M mutation), (2) activation of alternative pathways due to genetic alterations such as *MET* amplification and *KRAS* mutation, and (3) alteration of the phenotype, such as transformation to small cell lung cancer or EMT (epithelial-mesenchymal transition). Furthermore, these pathways may be co-activated in a single case, and their crosstalk can further complicate the cancer course and patient outcomes.

### 3.1. Mechanisms Underlying Development of Resistance to First- and Second-Generation EGFR-TKIs

Since first-generation EGFR-TKIs were introduced in clinical practice, the molecular mechanisms involved in EGFR-TKI acquired resistance have been vigorously investigated [37,38]. The most frequent mechanism associated with acquired resistance involves the T790M secondary mutation. Under physiological conditions, *EGFR*, a receptor-type tyrosine kinase, forms a dimer when bound by its ligands such as EGF and TGF-α and transmits pro-proliferation signals in the cells via autophosphorylation and assuming an activated conformation. In *EGFR* mutation-positive lung cancers, exon 19 deletion mutation and exon 21 L858R point mutation, among others, cause the kinase to become homeostatically activated in a ligand-independent manner [39]. This leads to the transmission of excessive pro-survival and pro-proliferation signals, which in turn lead to cancer initiation and progression; the efficacy of EGFR-TKIs lies in lowering these excessive pro-survival and pro-proliferation signals. When the T790M mutation occurs in the ATP-binding pocket of the kinase, the affinity of the kinase for EGFR-TKI decreases compared to that for ATP, leading to the development of drug resistance [40]. Generally, tumor heterogeneity in which multiple clones with different genotypes are present in one tumor mass has been observed [41]. It is known that in *EGFR* mutation-positive lung cancer, T790M-positive cells are present at an incidence rate of 79.9%, albeit at a low frequency when analyzed by an ultrasensitive method (0.009% to 26.9%), even before EGFR-TKI treatment [42]. It is hypothesized that these T790M-positive cells are protected from the effects of EGFR-TKI treatment and resistance acquired via the T790M mutation occurs frequently. Analysis using a highly sensitive method showed that the T790M mutation is present in 68% of cases of EGFR-TKI acquired resistance [43]. Osimertinib was developed based on the clear proof of concept that T790M was a potential target for therapeutic drug development and was successfully applied in clinical settings, even though it could not achieve a radical cure for NSCLC. Amplification of genes, such as *MET* and *ERBB2*, transformation to small cell lung cancer, hepatocyte growth factor (HGF) overexpression, EMT transition, and *PIK3CA* mutation/reduced PTEN expression are other events that can occur after treatment with first- and second-generation EGFR-TKIs and with cancer progression, leading to the development of drug resistance [44]. In addition, Akt is frequently activated in tumors after treatment with first- and second-generation EGFR-TKIs [45]. However, none of the drugs targeting these mechanisms have so far been applied in clinical settings.

### 3.2. Mechanisms Underlying Development of Resistance to Third-Generation EGFR-TKIs

The mechanisms underlying resistance to osimertinib, a third-generation EGFR-TKI, have been investigated [46,47], and the presence of the *EGFR* C797S mutation in addition to the T790M mutation (T790M/C797S double-positive) has been proposed as one of the main mechanisms responsible for resistance acquisition in T790M mutation-positive lung cancer [48]. C797S reduces the affinity between osimertinib and the *EGFR* kinase domain and increases the relative affinity for ATP. EGFR-TKIs that can inhibit the kinase activity of *EGFR* in T790M/C797S double-positive lung cancer have also been developed, but their clinical usefulness is yet to be approved. In the case of T790M loss, several genetic alterations have been observed, including those related to transformation to small cell lung cancer, *MET* amplification, *ERBB2* amplification/mutation, *PIK3CA* mutation, *KRAS* mutations, and oncogenic gene fusion involving *RET*, *FGFR3*, and *BRAF*.

The frequency and spectrum of resistance mechanisms observed when osimertinib is used as first-line therapy in T790M-negative patients are different from those in T790M-positive patients after second-line therapy [47]. The incidence of drug resistance acquisition has been increasing and becoming more complex in nature. In addition to the drug resistance conferred by the C797S mutation in *EGFR*, amplification of the MET gene as a bypass pathway, alteration in RAS-RAF-MAPK pathway, ROS1 fusion gene, PIK3CA mutation, and transformation to small cell lung cancer are other mechanisms that impart drug resistance. Thus, the resistance mechanisms acquired after osimertinib treatment are rather complex, and therefore, it is challenging to treat osimertinib resistance using a single therapeutic approach. Even if specific resistance factors are identified, there are currently no insurance-approved therapies to target these factors. At present, the most notable therapeutic strategy for addressing acquired resistance is the combination of amivantamab, a bispecific antibody targeting EGFR and MET, and lazertinib, a third-generation EGFR-TKI. In phase I clinical trial conducted in *EGFR* mutation-positive patients with disease progression after osimertinib, the overall response rate for both drugs was 36% [49]. A subgroup analysis of MET-based resistant cases revealed good results, with a response rate of 75%. Although HER3 alterations are not known to mediate resistance to EGFR-TKIs, HER3-targeted patritumab deruxtecan was used in phase I clinical trials in EGFR-TKI resistant cases and produced a response rate of 39%, which is a good result, regardless of the background resistance mechanism [50]. With respect to treatment after failure of 3rd generation EGFR-TKIs, there is an ongoing debate as to whether to target a specific resistance-forming mechanism or to promote the development of drugs that are widely indicated regardless of the resistance-forming mechanism. The use of next-generation sequencing to confirm the details of these genetic mutations in daily practice remains uncommon for several reasons, affecting the pace of resistance factor identification. *EGFR* mutation-positive lung cancer is a classic example of monoclonal evolution in tumors. Even within an individual, the genomic landscape of individual tumors is heterogeneous, and the therapeutic agent of choice reveals the predominant subclones that are least susceptible to treatment over time [51]. Liquid biopsies have enabled the monitoring of genomic changes by analyzing circulating tumor (ct) DNA. Several clinical trials have reported that clearance of *EGFR*-activated mutations several weeks after the start of treatment is associated with good PFS, and liquid biopsy results have also become predictive markers for efficacy and residual lesions [52,53,54,55]. If the measurement of plasma ct DNA can be popularized in general practice, it would be possible to select treatments based on resistance status more widely (Figure 1).

## 4. Possibility of Combination Therapy with EGFR-TKI and Other Drugs

For curative treatment, one strategy for drastically reducing the number of remaining tumor cells and preventing the development of drug resistance is to combine EGFR-TKI with other therapeutic agents. Based on this concept, combination therapy, including EGFR-TKIs and cytotoxic anticancer agents or angiogenesis inhibitors, is being explored. The usefulness of combination therapy is positioned as a therapeutic strategy aimed at curative treatment in the sense that cancer cells can be killed regardless of the genetic background, taking into account that tumor heterogeneity comprises a population of cells with various genetic backgrounds. In fact, some combination regimens have been shown to prolong the duration of response compared to TKI monotherapy, but it is still controversial whether these strategies increase the cure rate.

### 4.1. Combination Therapy with Cytotoxic Anticancer Agents

Although EGFR-TKIs are the key drug for treating *EGFR* mutation-positive lung cancer, cytotoxic anticancer agents remain an important treatment option. Cytotoxic anticancer agents attack cancer cells with or without *EGFR* mutations, and their combination with EGFR-TKIs may prevent the growth of EGFR-TKI-resistant cells. A next-generation sequencing-based study revealed that very few T790M-positive cells exist among *EGFR* mutation-positive cells at the early stage of cancer before the treatment is started [56]. Thus, it is prudent to attack these cells with cytotoxic anticancer agents from the very beginning, since T790M acquired resistance is expected to occur in *EGFR* T790M mutation-positive cells, which were initially very few. Moreover, cells without T790M mutation may escape cell death by activating bypass pathways of survival and proliferation, other than those mediated by *EGFR*, upon exposure to EGFR-TKIs. Therefore, combination therapy with cytotoxic anticancer agents is expected to be more effective in such cases.

For combination anticancer therapeutic approaches, the use of carboplatin and PEM is recommended based on the results of clinical trials [20,21]. However, with the combined use of cytotoxic chemotherapy, treatment-related adverse events of Common Terminology Criteria for Adverse Events (CTCAE) grade 3 or higher increased to 65.3% (31.0% for TKI monotherapy). In particular, grade 3 or higher neutropenia was as high as 31.2% with combination therapy, compared to 0.6% with TKI monotherapy [20]. The changes in frequency and spectrum of the resistance mechanism in response to combination therapy using PEM and EGFR-TKI remain undetermined and should be explored in future studies. In vitro, PEM has been reported to regulate the PI3K/Akt pathway downstream of *EGFR* [57,58]. We previously reported that PTK2/FAK activation may occur in cells that acquire drug resistance in response to PEM and EGFR-TKI treatment [59]. Thus, the cells treated with combination therapy may activate combination signaling pathways including those activated in response to targeted and non-targeted drugs, thereby leading to the emergence of a novel resistance mechanism. We are currently investigating such pathways using cells that have acquired resistance to the combination therapy. In the future, it will be necessary to evaluate whether the simultaneous or sequential use of two drugs is more likely to improve prognosis from the perspective of resistance mechanisms.

### 4.2. Combination Therapy with Angiogenesis Inhibitors

Scientific studies have shown that aspects of the tumor microenvironment, such as survival signals from fibroblasts around cancer cells and poor vascularization around cancer cells, may cause cancer cells to survive EGFR-TKI monotherapy [60]. In fact, combination therapy with angiogenesis inhibitors and EGFR-TKIs, which is believed to inhibit neovascularization, has been shown to prolong PFS in several clinical trials [22,23]. However, there is no clear evidence that it increases the cure rate; moreover, the RELAY study [24], which compared erlotinib monotherapy with erlotinib plus ramucirumab combination therapy, reported no difference in the detection rate of T790M during disease progression. With the combined use of angiogenesis inhibitors, hypertension of CTCAE grade 3 or higher increased to 23% (compared to 1% with TKI monotherapy) [24]. Whether the combination of angiogenesis inhibitors and EGFR-TKIs modifies the resistance mechanism is a subject for further investigation. The mechanisms of resistance to angiogenesis inhibitors also need to be studied in more detail [61,62]. Furthermore, a phase II trial comparing single-agent osimertinib with the combination of osimertinib plus bevacizumab in patients with the T790M mutation [63] showed no benefit in prolonging PFS in the combination group. Therefore, it may be necessary to identify a biomarker to stratify patients who can benefit from the combination therapy using angiogenesis inhibitors.

## 5. Conclusions

EGFR-TKIs remain the mainstay of pharmacotherapy for the treatment of *EGFR* mutation-positive lung cancer. However, recurrence due to cancer heterogeneity and mechanisms underlying drug resistance remain persistent challenges. Moreover, it is challenging to achieve a radical cure using a single agent. On the contrary, remarkable progress has been made in elucidating the mechanisms underlying the development of resistance to molecularly targeted drugs, and it will not be long before therapeutic strategies based on the results of scientific studies will be available in clinical practice. To determine the optimal approach, we need to design a therapeutic strategy to overcome the resistance mechanism in cells. The future challenge is to clarify the key factors associated with drug resistance and the combinations of drugs that may be clinically more effective by suppressing the EGFR pathway.

## Figures and Tables

**Figure 1 cells-10-03192-f001:**
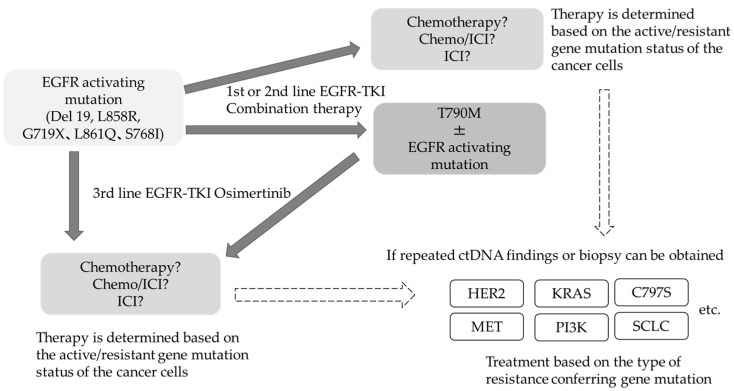
Future prospects of treatment algorithms for *EGFR* mutation-positive lung cancer.

**Table 1 cells-10-03192-t001:** List of clinical trials evaluating EGFR-TKI monotherapy in EGFR-mutated lung cancer.

Trial (Reference)	Target EGFR Mutation	Experimental Therapy	Control Therapy	PFS ^1^ (Primary Outcome)			Overall Survival		
				Months		HR ^4^ (95% C.I. ^5^)	Months		HR (95% C.I.)
				Exp ^2^	Ctrl ^3^		Exp	Ctrl	
NEJ002(2)	Sensitizing mutation	Gefitinib	Cb ^6^/PTX	10.8	5.4	0.33	30.5	23.6	-
						(0.22–0.41)			
WJTOG 3405(3)	Ex19_del Ex21_L858R	Gefitinib	CDDP ^7^/DTX ^8^	9.2	6.3	0.489	30.9	NR ^9^	1.638
						(0.336–0.710)			(0.749–3.582)
EURTAC(13)	Ex19_del Ex21_L858R	Erlotinib	CDDP	9.7	5.2		-	-	-
EURTAC(13)	Ex19_del Ex21_L858R	Erlotinib	Cb/DTX	9.7	5.2	0.37	-	-	-
			GEM ^10^			(0.25–0.54)			
OPTIMAL (14)	Ex19_del Ex21_L858R	Erlotinib	Cb/GEM	13.1	4.6	0.16	-	-	-
OPTIMAL (14)	Ex19_del Ex21_L858R	Erlotinib	Cb/GEM	13.1	4.6	(0.10–0.26)	-	-	-
LUX-Lung 7 (15)	Ex19_del Ex21_L858R	Afatinib	Gefitinib	11	10.9	0.74	27.9	24.5	0.86
LUX-Lung 7 (15)	Ex19_del Ex21_L858R	Afatinib	Gefitinib	11	10.9	(0.57–0.95)	27.9	24.5	(0.66–1.12)
ARCHER 1050 (16)	Ex19_del Ex21_L858R	Dacomitinib	Gefitinib	14.7	9.2	0.59	34.1	26.8	0.76
ARCHER 1050 (16)	Ex19_del Ex21_L858R	Dacomitinib	Gefitinib	14.7	9.2	(0.47–0.74)	34.1	26.8	(0.582–0.993)
FLAURA (17)	Ex19_del Ex21_L858R	Osimertinib	Gefitinib	18.9	10.2	0.46	38.6	31.8	0.799
FLAURA (17)	Ex19_del Ex21_L858R	Osimertinib	Erlotinib	18.9	10.2	(0.37–0.57)	38.6	31.8	(0.641–0.997)

^1^ PFS, progression free survival; ^2^ Exp, Experimental therapy; ^3^ Ctrl, Control therapy; ^4^ HR, hazard ratio; ^5^ C.I., confidence interval; ^6^ Cb, carboplatin; ^7^ CDDP, cisplatin; ^8^ DTX, docetaxel; ^9^ NR, not reached; ^10^ GEM, gemcitabine.

**Table 2 cells-10-03192-t002:** List of clinical trials evaluating EGFR-TKI combination therapy in EGFR-mutated lung cancer.

Trial (Reference)	Target EGFR Mutation	Experimental Therapy	Control Therapy	PFS ^1^ (Primary Outcome ^†^)			Overall Survival		
				Months		HR ^4^ (95% C.I. ^5^)	Months		HR (95% C.I.)
				Exp ^2^	Ctrl ^3^		Exp	Ctrl	
NEJ009 (20)	Sensitizing mutation	Cb ^6^/PEM ^7^/ Gefitinib	Gefitinib	20.9	11.9	0.49	50.9	28.8	0.72
						(0.39–0.62)			(0.55–0.95)
Noronha trial (21)	Sensitizing mutation	Cb/PEM/ Gefitinib	Gefitinib	16.0	8.0	0.51	NR ^8^	17	0.45
						(0.39–0.66)			(0.31–0.65)
NEJ026 (22,23)	Ex19_del Ex21_L858R	Erlotinib/ Bevacizumab	Erlotinib	16.9	13.3	0.605	50.7	46.2	1.007
						(0.417–0.877)			(0.681–1.49)
RELAY (24)	Ex19_del Ex21_L858R	Erlotinib/ Bevacizumab	Erlotinib	19.4	12.4	0.59	NR	NR	0.83
						(0.46–0.76)			(0.53–1.30)

^1^ PFS, progression free survival; ^2^ Exp, Experimental therapy; ^3^ Ctrl, Control therapy; ^4^ HR, hazard ratio; ^5^ C.I., confidence interval; ^6^ Cb, carboplatin; ^7^ PEM, pemetrexed; ^8^ NR, not reached.^†^ NEJ009 has primary outcomes (PFS, PFS2 and overall survival) using predetermined hierarchical sequential testing method.

## Data Availability

Not applicable.

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
