# Peer review of "Current Therapeutic Strategies and Prospects for EGFR Mutation-Positive Lung Cancer Based on the Mechanisms Underlying Drug Resistance"

_cells, 2021, doi:10.3390/cells10113192_

Round 1

Reviewer 1 Report

The authors addressed the topic of target therapies and mechanisms underlying
drug resistance for lung cancers with EGFR mutations by providing precise
and detailed information.
The only suggestion I can give is to make table 2 clearer,
as I found it difficult to understand

Author Response

Dear Dr. Patty Wang:

I wish to submit the revised version of our manuscript titled “Current therapeutic strategies and prospects for EGFR mutation-positive lung cancer based on the mechanisms underlying drug resistance” for publication in Cells. The manuscript ID is “cells-1406155”.

We thank you and the reviewers for reviewing our manuscript and for providing helpful suggestions. Our point-by-point responses to the reviewers’ comments are provided below, with a description of the changes made in the manuscript.

We hope that we have adequately addressed the reviewers’ concerns and that the manuscript is now suitable for publication.

Thank you for your consideration. We look forward to hearing from you.

Sincerely,

Yukari Tsubata, M.D., Ph.D.

Department of Internal Medicine, Division of Medical Oncology & Respiratory Medicine, Shimane University Faculty of Medicine

Address: 89-1 Enya-cho, Izumo, Shimane 693-8501, Japan

Tel/Fax: +81-853-20-2580

Responses to Reviewer 1

We appreciate your helpful comments and suggestions. The formats of Tables 1 and 2 have been changed to improve clarity and readability.

Reviewer 2 Report

This manuscript is about  review of EGFR-TKI resistant mechanisms. Overall a well-written manuscript, but it needs some corrections.   

Line34-36: adding references related to the content is required.

Line83: Please add sentences about accurate clinical treatments.

Line104: Please add references about combination therapy with an agent (especially closely related to EGFR-TKI resistance).  And it would be better if the progress of the clinical trials to date is added.

Line169: It would be better to add information about the onset portion of C797S among EGFR mutations.

Line177-180: It would be better if you add the references and the rationale for the genetic alteration that appears of T790M.

Line198: The contents of the clinical trial updated to date should be summarized in a Table which leads the manuscript quality to be much better.

Line 215: Please add the side effects expected when combination therapy with a cytotoxic anticancer agent are performed and add the kinds of cytotoxic anticancer agents which can be used as combination therapy.

Line247: Authors described the use of anticancer agents in combination therapy for overcoming EGFR-TKI's resistance. In this case, what kinds of mechanisms are involved ? And what are the advantages over combination therapy ? It would be nice if there was some comments on whether there is an expected way to further increase the treatment efficiency.

Figure 1: It is necessary to confirm whether osimertinib is treated even when T790M is negative.          

Author Response

Responses to Reviewer 2

We appreciate your helpful comments and suggestions. The manuscript has been revised based on your comments, as follows:

Line34-36: adding references related to the content is required.

We have added references 4-8 and corrected the numbering of the reference list accordingly.

Line83: Please add sentences about accurate clinical treatments.

Thank you for this feedback. We have added the following sentence to this section:

“Based on the results of these pivotal clinical trials, both first- and second-generation EGFR-TKIs are being used in clinical settings world-wide.”

Line104: Please add references about combination therapy with an agent (especially closely related to EGFR-TKI resistance).  And it would be better if the progress of the clinical trials to date is added.

We appreciate your helpful comments and suggestions. To the best of our understanding, there is no treatment regimen currently that opens the door to clinical application. The following sentences has been added to this section regarding the results of clinical trials evaluating the effectiveness of combination therapy with third generation EGFR-TKIs.

“Meanwhile, several clinical trials have begun exploring the usefulness of osimertinib combination therapy. Combination therapy of osimertinib and CBDCA / PEM in FLAURA2, an international safety run study, exhibited a manageable safety and toler-ability profile in first line settings [18]. In addition, in a phase II clinical trial (WJOG9717L) examining the efficacy of bevacizumab (an angiogenesis inhibitor) and osimertinib, the primary endpoint, PFS, did not show a statistically significant prolongation of survival [19]. Further assessment is expected.“

Line169: It would be better to add information about the onset portion of C797S among EGFR mutations.

The C797S mutation is one of the acquired mechanisms of resistance that has been observed after administration of 3rd generation EGFR-TKIs. We have mentioned the resistance mechanism of C797S in Section 3.2.

Line177-180: It would be better if you add the references and the rationale for the genetic alteration that appears of T790M.

Thank you for your advice. We have added the following sentences to this section”:

“Generally, tumor heterogeneity in which multiple clones with different genotypes are present in one tumor mass has been observed [41]. It is known that in EGFR mutation-positive lung cancer, T790M-positive cells are present at an incidence rate of 79.9%, albeit at a low frequency when analyzed by an ultrasensitive method (0.009% to 26.9%), even before EGFR-TKI treatment [42]. It is hypothesized that these T790M-positive cells are protected from the effects of EGFR-TKI treatment and resistance acquired via the T790M mutation occurs frequently. “

Line198: The contents of the clinical trial updated to date should be summarized in a Table which leads the manuscript quality to be much better.

We appreciate your helpful comments and suggestions.

The mechanisms of acquired resistance after administration of third-generation EGFR-TKIs are diverse, and there are innumerable in vitro and in vivo studies are being performed to examine the targeting of each molecule. For example, one trial attempted to combine a third-generation EGFR-TKI with a MEK inhibitor from first-line treatment for the purpose of preventing the emergence of resistance (Tricker EM, et al Cancer Discov 5 (9): 960–971, 2015); while another clinical trial focused on C797S (Uchibori K, et al Nat Commun 13; 8: 14768, 2017), investigating the efficacy of brigatinib combined with an anti-EGFR antibody for overcoming resistance. Although clinical trials have been conducted on many target molecules, the clinical development of treatments for acquired resistance has stalled, and no established treatment is available at this time. Due to the large number of research reports, instead of creating a table, we have noted the latest clinical trial data that may lead to clinical practice, as follows:

“At present, the most notable therapeutic strategy for addressing acquired resistance is the combination of amivantamab, a bispecific antibody targeting EGFR and MET, and lazertinib, a third-generation EGFR-TKI. In a phase I clinical trial conducted in EGFR mutation-positive patients with disease progression after osimertinib, the overall response rate for both drugs was 36% [49]. A subgroup analysis of MET-based resistant cases revealed good results, with a response rate of 75%. Although HER3 alterations are not known to mediate resistance to EGFR-TKIs, HER3-targeted patritumab deruxtecan was used in phase I clinical trials in EGFR-TKI resistant cases and produced a response rate of 39%, which is a good result, regardless of the background resistance mechanism [50]. With respect to treatment after failure of 3rd generation EGFR-TKIs, there is an ongoing debate as to whether to target a specific resistance-forming mechanism or to promote the development of drugs that are widely indicated regardless of the resistance-forming mechanism.” 

Line 215: Please add the side effects expected when combination therapy with a cytotoxic anticancer agent are performed and add the kinds of cytotoxic anticancer agents which can be used as combination therapy.

Thank you for your feedback. We have added the following sentences to this section:

“However, with the combined use of cytotoxic chemotherapy, treatment-related adverse events of Common Terminology Criteria for Adverse Events (CTCAE) grade 3 or higher increased to 65.3% (31.0% for TKI monotherapy). In particular, grade 3 or higher neutropenia was as high as 31.2% with combination therapy, compared to 0.6% with TKI monotherapy [20].”

“With the combined use of angiogenesis inhibitors, hypertension of CTCAE grade 3 or higher increased to 23% (compared to 1% with TKI monotherapy) [24].”

Line247: Authors described the use of anticancer agents in combination therapy for overcoming EGFR-TKI's resistance. In this case, what kinds of mechanisms are involved ? And what are the advantages over combination therapy ? It would be nice if there was some comments on whether there is an expected way to further increase the treatment efficiency.

We appreciate your helpful comments and suggestions. We have added the following sentences to this section:

“The usefulness of combination therapy is positioned as a therapeutic strategy aimed at curative treatment in the sense that cancer cells can be killed regardless of the genetic background, taking into account that tumor heterogeneity comprises a population of cells with various genetic backgrounds. In fact, there are some combination regimens that have been shown to prolong the duration of response compared to TKI monotherapy, but it is still controversial whether these strategies increase the cure rate.”

Figure 1: It is necessary to confirm whether osimertinib is treated even when T790M is negative.

Thank you for your helpful comments. We have revised Figure 1 accordingly.

Reviewer 3 Report

Non-small cell lung cancer (NSCLC) represents a tumor entity perfectly illustrating the significant therapeutic progress made in oncology by the introduction of targeted therapeutics. In this context, the development of several generations of EGFR inhibitors and the emergence of therapy-associated EGFR mutations in NSCLC is a remarkable example of co-evolution. 

In their review, Tsubata et al. describe 1st, 2nd and 3rd generation EGFR inhibitors with respect to clinical outcome data in NSCLC and the occurrence of resistance. The manuscript is clearly structured and nicely written. However, in my opinion, the review is not sufficiently detailed to provide an extra benefit against the background of nummerous reviews dealing with clinical effects of EGFR inhibitors in NSCLC and resistance factors  involved in treatment failure. For example, the three years old review of Nagano et al. published in Cells (PMID: 30445769) is in my view more informative than the actual manuscript regarding mechanisms critically involved in resistance against EGFR inhibitors in NSCLC. Likewise, I feel that the review of Wu & Shi (PMID: 29455650) from 2018 provides more insights into potential therapeutic strategies dealing with EGFR inhibitor resistance (continuous therapy, vertical or horizontal pathway inhibition, immune therapies). 

To summarize, I would suggest that the actual manuscript should be substantially extended. Aspects that could be incorporated are, e.g, the concept of acquired vulnerability, the role of other RTKs (besides MET) or RTK ligands (besides HGF), the aspect of squamous transformation in osimertinib resistance, the role of liquid biopsies in biomonitoring  mutational changes, the delineation of targeted therapeutics which address signaling molecules downstream of RTKs, or the interplay of EGFR inhibition and immune system (effect on PD1/PD-L1 pathway, effect on secretion of inflammatory mediators). 

I think the different facets of therapy resistance against EGFR inhibitors in NSCLC still justify an up-to-date review. However - at least in my opinion -  the manuscript in the present form does not satisfactorily reflect the actual developments. 

Author Response

Responses to Reviewer 3

We appreciate your helpful comments and suggestions. The reviews by Nagano et al. and Wu and Shih were very informative and helpful. As you have noted, in the field of EGFR mutation-positive lung cancer treatment, gene mutation analysis from ctDNA has been introduced in daily practice over the last few years, and research on interactions with the PD-1 / L1 pathway has made significant advances and clear progress in elucidating the treatment sequence. Our review has focused on discussing treatment strategies based on gene mutation results. From this perspective, we introduce the latest information in the field and focus on treatment sequences that are expected to be clinically applied in the near future. We have clarified this point, and added further analysis, as suggested.

We have revised the introduction as follows:

“This review focuses specifically on the latest treatments for EGFR mutation-positive lung cancer and the strategies aimed at overcoming drug resistance, ..”

We have added a new section and the following sentences to the main text:

Line 280:

“EGFR mutation-positive lung cancer is a classic example of monoclonal evolution in tumors. Even within an individual, the genomic landscape of individual tumors is heterogeneous, and the therapeutic agent of choice reveals the predominant subclones that are least susceptible to treatment over time [51]. Liquid biopsies have enabled the monitoring of genomic changes by analyzing circulating tumor (ct) DNA. Several clinical trials have reported that clearance of EGFR-activated mutations several weeks after the start of treatment is associated with good PFS, and liquid biopsy results have also become predictive markers for efficacy and residual lesions [52-55].”

Line166:

“2.5 Immunotherapy for EGFR-mutated lung cancer

 These days, immune checkpoint inhibitors (ICIs) (including anti-PD-1 / L1 and anti-CTLA-4 antibodies) are an essential therapeutic option in the treatment of lung cancer. EGFR activation may increase PD-L1 expression in tumor cells by upregulating PI3K and activating the IL-6 / JAK / STAT3 pathway and MEK / ERK pathway [25-28]. It has also been reported that administration of EGFR-TKIs has a positive effect on the tumor microenvironment [29-31], but the specific molecules involved in PD-L1 activation have not yet been elucidated. Furthermore, the correlation between the presence of EGFR mutations and PD-L1 expression is still controversial. However, the intensity and treatment susceptibility of PD-L1 expression in T790M-negative patients are consistently correlated, regardless of prior treatment with EGFR-TKIs [32,33]. This indicates that elucidating the relationship between PD-L1 expression and EGFR-TKI susceptibility, as well as the resistance mechanisms, may provide new avenues for overcoming EGFR-TKI resistance.

On the other hand, the therapeutic effect of combining ICIs and chemotherapy for treatment after EGFR-TKI failure remains unclear because EGFR-positive lung cancer has been excluded from many clinical trials [34]. The IMpower150 study group examined the effect of adding atezolizumab to CBDCA / PTX / BEV in chemotherapy naive non-squamous NSCLC and allowed registration of EGFR-positive lung cancer after EGFR-TKI failure. A subgroup analysis of this cohort revealed useful results for median OS in the atezolizumab-added group (not evaluated vs 17.5 months, HR 0.31) [35]. However, the therapeutic effect of ICIs alone for EGFR-positive lung cancer is limited, and the PFS of nivolumab monotherapy after EGFR-TKI failure is only 1.7 months [36]. It is presumed that such a gap between the results of preclinical studies and actual clinical trials is due to various factors, such as tumor heterogeneity, PD-L1 IHC scoring criteria, and treatment history. Because some EGFR mutation-positive patients may benefit from immunotherapy, it is important to elucidate the molecules primarily involved in the mechanism of PD-L1 activation and EGFR-TKI resistance, and to investigate their association. Identifying predictors of therapeutic efficacy is considered to be an important element of personalized treatment.”